

# Four tyrosine residues of the rice immune receptor XA21 are not required for interaction with the co-receptor OsSERK2 or resistance to *Xanthomonas oryzae* pv. *oryzae*

Daniel F. Caddell[1,2], Tong Wei[1,3], Sweta Sharma[1], Man-Ho Oh[4,5], Chang-Jin Park[1,6], Patrick Canlas[1], Steven C. Huber[4,7] and Pamela C. Ronald[1]

[1] Department of Plant Biology and the Genome Center, University of California, Davis, CA, USA
[2] Current affiliation: Department of Plant and Microbial Biology, University of California, Berkeley, CA, USA
[3] Current affiliation: BGI-Shenzhen, Shenzhen, China
[4] Department of Plant Biology, University of Illinois at Urbana-Champaign, Urbana, IL, USA
[5] Current affiliation: Department of Biological Science, Chungnam National University, Daejeon, South Korea
[6] Current affiliation: Department of Bioresources and Engineering and PERI, Sejong University, Seoul, South Korea
[7] Agricultural Research Service, United States Department of Agriculture, Urbana, IL, USA

Corresponding author
Pamela C. Ronald,
pcronald@ucdavis.edu

## ABSTRACT

Tyrosine phosphorylation has emerged as an important regulator of plasma membrane-localized immune receptors activity. Here, we investigate the role of tyrosine phosphorylation in the regulation of rice *XANTHOMONAS* RESISTANCE 21 (XA21)-mediated immunity. We demonstrate that the juxtamembrane and kinase domain of *Escherichia coli*–expressed XA21 (XA21JK) autophosphorylates on tyrosine residues. Directed mutagenesis of four out of the nine tyrosine residues in XA21JK reduced autophosphorylation. These sites include $Tyr^{698}$ in the juxtamembrane domain, and $Tyr^{786}$, $Tyr^{907}$, and $Tyr^{909}$ in the kinase domain. Rice plants expressing XA21-GFP fusion proteins or proteins with these tyrosine residues individually mutated to phenylalanine ($XA21^{YF}$-GFP), which prevents phosphorylation at these sites, maintain resistance to *Xanthomonas oryzae* pv. *oryzae*. In contrast, plants expressing phosphomimetic XA21 variants with tyrosine mutated to aspartate ($XA21^{YD}$-GFP) were susceptible. In vitro purified $XA21JK^{Y698F}$, $XA21JK^{Y907F}$, and $XA21JK^{Y909F}$ variants are catalytically active, whereas activity was not detected in $XA21JK^{Y768F}$ and the four $XA21JK^{YD}$ variants. We previously demonstrated that interaction of XA21 with the co-receptor OsSERK2 is critical for biological function. Four of the $XA21JK^{YF}$ variants maintain interaction with OsSERK2 as well as the XA21 binding (XB) proteins XB3 and XB15 in yeast, suggesting that these four tyrosine residues are not required for their interaction. Taken together, these results suggest that XA21 is capable of tyrosine autophosphorylation, but the identified tyrosine residues are not required for activation of XA21-mediated immunity or interaction with predicted XA21 signaling proteins.

*Xanthomonas oryzae* pv. *oryzae*, Rice

## INTRODUCTION

Plasma membrane-localized immune receptors initiate plant defense responses upon recognition of conserved microbial molecules. Three well-characterized plasma membrane-localized immune receptors are Arabidopsis FLAGELLIN-SENSING 2 (FLS2), Arabidopsis EF-Tu RECEPTOR (EFR), and rice *XANTHOMONAS* RESISTANCE 21 (XA21). FLS2, EFR, and XA21 recognize bacterial flagellin (*Gomez-Gomez & Boller, 2000*), bacterial elongation factor-Tu (*Zipfel et al., 2006*), and the *Xanthomonas oryzae* pv. *oryzae* (*Xoo*) peptide Required for activation of XA21-mediated immunity X (RaxX) (*Pruitt et al., 2015*), respectively. Immune activation of FLS2, EFR, and XA21 requires interaction with members of the SOMATIC EMBRYOGENESIS RECEPTOR-LIKE KINASE (SERK) family of proteins. In Arabidopsis, EFR and FLS2 interact with AtSERK3, commonly referred to as BRI1-ASSOCIATED KINASE1 (BAK1) (*Chinchilla et al., 2007*). Recognition of the cognate bacterial peptide induces direct heterodimerization and transphosphorylation between BAK1 and its partner receptor (*Chinchilla et al., 2007*). In rice, XA21 and the immune receptors OsFLS2 (*Takai et al., 2008*) and XA3 (*Xiang et al., 2006*) require OsSERK2 for biological function (*Chen et al., 2014*). The interaction and transphosphorylation between XA21 and OsSERK2 occurs in the absence of *Xoo* (*Chen et al., 2014*).

 *XANTHOMONAS* RESISTANCE 21-mediated immunity is regulated through post-translational modifications and interactions with XA21 BINDING PROTEINS (XBs). For example, XB3, an E3-ubiquitin ligase, is activated by XA21 and is predicted to assist in XA21-mediated immunity through ubiquitination of proteins (*Wang et al., 2006*). XB15, a protein phosphatase, dephosphorylates XA21 residues to attenuate XA21-mediated immunity (*Park et al., 2008*). XB24, an ATPase, maintains XA21 in an inactive state by promoting autophosphorylation of XA21 residues. Bacterial recognition promotes disassociation of XB24 from XA21, leading to kinase activation (*Chen et al., 2010b*). Although specific phosphosites are presumed to be involved that activate or inhibit XA21 kinase activity, they are largely unknown. XA21 kinase activity is required for full functionality. Rice plants expressing a catalytically inactive XA21 carrying a glutamate mutation in the conserved lysine residue of the XA21 catalytic domain (XA21$^{K736E}$) are partially resistant to *Xoo* (*Andaya & Ronald, 2003*). Another XA21 variant, which has the aspartic acid residue required for phospho-transfer (*Nolen, Taylor & Ghosh, 2004*) swapped to asparagine (XA21$^{D842N}$), is also catalytically inactive (*Chen et al., 2014*). Phosphorylation of serine and threonine residues in the juxtamembrane (JM) domain of XA21 regulate protein stability (*Xu et al., 2006*), kinase activity (*Chen et al., 2010a*), and protein–protein interactions (*Chen et al., 2010a*; *Park et al., 2008*).

 While serine and threonine are the major sites of phosphorylation in most plant proteins (*Nakagami et al., 2010*), tyrosine is also an important residue for regulating plant growth, development, and immunity (*Macho, Lozano-Duran & Zipfel, 2015*). To study
tyrosine phosphorylation, tyrosine residues can be modified using site-directed mutagenesis to phenylalanine (Y-to-F, or "YF" in this paper) or aspartate (Y-to-D, or "YD" in this paper), which prevents phosphorylation or serves to mimic the negative charge of a phosphorylated tyrosine (phosphomimetic), respectively. For example, the Arabidopsis leucine-rich repeat receptor-like kinase (LRR-RLK) BRASSINOSTEROID INSENSITIVE 1 (BRI1), which recognizes the growth hormone brassinosteroid, is phosphorylated on Tyr$^{831}$ and Tyr$^{956}$ in planta. BRI1 variants that carry Tyr$^{831}$ replaced with phenylalanine (BRI1$^{Y831F}$) retain kinase activity but display increased overall growth with altered leaf shape and reduced flowering time when expressed in the *bri1-5* mutant background (*Oh et al., 2009*). In contrast, the BRI1 variant with Tyr$^{956}$ modified to phenylalanine (BRI1$^{Y956F}$) is catalytically inactive (*Oh et al., 2009*). Phosphorylation of Tyr$^{956}$ is inhibitory rather than essential for activity (*Oh, Clouse & Huber, 2012*), consistent with results obtained with the *Arachis* Symbiosis Receptor Kinase, AhSYMRK (*Paul et al., 2014*), where autophosphorylation of the homologous tyrosine residue on AhSYMRK (Tyr$^{670}$) also inhibited catalytic activity. Collectively, these results suggest that tyrosine phosphorylation is critical for BRI1-mediated growth and development.

Tyrosine phosphorylation also serves an important role in EFR-mediated immunity. Phosphorylation of Tyr$^{836}$ is necessary for activating EFR-mediated immunity. EFR$^{Y836F}$ mutants are capable of elf18-induced dimerization with BAK1, but fail to initiate an EFR-mediated immune response. As a means to incapacitate EFR-mediated immunity, phytopathogenic *Pseudomonas syringae* secretes a tyrosine phosphatase, HopAO1, that dephosphorylates phospho-Tyr$^{836}$ (*Macho et al., 2014*). The role of tyrosine phosphorylation was also assessed in the Arabidopsis LysM-RLK CHITIN ELICITOR RECEPTOR KINASE 1 (CERK1). Similar to EFR, autophosphorylation of the analogous tyrosine residue, Tyr$^{428}$, is required for in vivo activation (*Suzuki et al., 2018*) and chitin-triggered CERK1 activation (*Liu et al., 2018*). Dephosphorylation of Tyr$^{428}$ by the CERK1-interacting protein phosphatase 1 reduces CERK1 signaling (*Liu et al., 2018*).

While the significance of tyrosine phosphorylation regulating LRR-RLKs in Arabidopsis and *Arachis* has been demonstrated for some proteins, its role has not been studied in other species. In this study, we investigated the role of tyrosine phosphorylation of rice XA21. We found that *Escherichia coli*–expressed XA21 autophosphorylates on tyrosine residues, and that Tyr$^{698}$, Tyr$^{786}$, Tyr$^{907}$, and Tyr$^{909}$ are potential autophosphorylation sites. However, transgenic rice expressing XA21 variants with these tyrosine residues replaced with phenylalanine (XA21$^{YF}$) maintain resistance to *Xoo* and interaction with XBs, suggesting that XA21 is capable of tyrosine autophosphorylation, but the identified tyrosine residues are not required for activation of XA21-mediated immunity or interaction with predicted XA21 signaling proteins.

## MATERIALS AND METHODS

### Site-directed mutagenesis and plasmid construction

A previously generated GST-XA21JK construct (*Liu et al., 2002*) was used as the template for site-directed mutagenesis using the QuikChange XL site-directed mutagenesis kit (Stratagene, San Diego, CA, USA). Individual constructs were generated with the following

substitutions: Y698F, Y723F, Y786F, Y829F, Y889F, Y894F, Y907F, Y909F, and Y936F. GST-XA21JK variants K736E, S697A, S697D, T705A, T705E, T680A, S699A were generated previously (*Chen et al., 2010a*). XA21JK/pENTR (*Chen et al., 2014*) was used as the template for site-directed mutagenesis of Y698F, Y698D, Y786F, Y786D, Y907F, Y907D, Y909F, and Y909D. Gateway cloning using Gateway LR Clonase (Invitrogen, Carlsbad, CA, USA) was utilized to recombine XA21JK/pENTR construct and mutant variants into a His-Nus-fusion vector (*Chern et al., 2005*; *Schwessinger et al., 2011*) for in vitro kinase autophosphorylation assay. His-Nus-XA21JK$^{D842N}$ was previously generated (*Chen et al., 2014*). The pNlexA-XA21JK vectors used for yeast analyses were similarly constructed by recombining the XA21JK/pENTR constructs into pNlexAgtwy (a gateway compatible vector modified from pNLexA (Clontech, Mountain View, CA, USA) (*Chen et al., 2010b*). LexA-GUS, LexA-XA21JK$^{D842N}$, HA-XB3, HA-XB15, and HA-OsSERK2JMK vectors were generated previously (*Chen et al., 2010a*, *2014*). XA21-GFP/pC1300 (*Park et al., 2010*) was used as template for site-directed mutagenesis of Y698F, Y698D, Y786F, Y786D, Y907F, Y907D, Y909F, and Y909D for rice studies. Point mutations in the appropriate vectors were introduced as described above. All constructs were sequenced in both directions to verify specific mutations and lack of additional mutations.

## Rice transformation

Rice transformations were performed as described previously (*Chern et al., 2005*). *Agrobacterium* strain EHA105 was used to infect calli generated from the Kitaake rice cultivar. Transformants were selected using hygromycin as a selectable marker and later confirmed by PCR using transgene specific primers (Table S1).

## Plant growth conditions and inoculations assays

Rice plants (*Oryza sativa* ssp. *japonica* L., cultivar Kitaake) were grown in the greenhouse for 6 weeks and transferred to growth chambers for inoculation with 14h day/10 h night cycle, 28/26 °C temperature, and 85–95% humidity. *Xoo* Philippine race 6 strain PXO99Az (PXO99 in this paper), was cultured on PSA media containing cephalexin for 2–3 days and suspended in sterile water to a final $OD_{600}$ of 0.5. Healthy and fully expanded leaves of each plant were inoculated using the scissors inoculation method that has been established previously (*Chern et al., 2005*). Infection was measured 14 days later.

## Protein extraction

Approximately seven cm (0.05 g) of tissue from fully expanded rice leaves were crushed to a fine powder in liquid nitrogen. Samples were incubated in rice extraction buffer (two mM EDTA, 0.15M NaCl, 0.01M sodium phosphate, pH 7.2, 1% (v/v) Triton X-100, 10 mM ß-Me, one mM PMSF, 1% (v/v) sigma protease inhibitor cocktail P-9599) for 30 min on ice. Samples were cleared by centrifugation at 13,000 rpm, 4 °C, for 10 min. Protein was denatured in $4\times$ SDS loading buffer and incubated at 80 °C for 10 min.

## Gene expression assays

Gene expression assays were performed as previously described (*Pruitt et al., 2015*). Rice plants were grown hydroponically and watered twice a week with Hoagland solution.

About two-cm leaf strips from 4-week-old plants were floated in sterile water overnight. The leaf tissues were treated for 6 h in the second morning with water, one µM nonsulfated or sulfated 21-amino acid RaxX peptides (RaxX21-Y or RaxX21-sY, respectively; Pacific Immunology, Ramona, CA, USA), and immediately frozen in liquid nitrogen for RNA extraction. Total RNA was isolated with TRIzol reagent (Invitrogen, Carlsbad, CA, USA), treated with DNase I, and used for cDNA synthesis with Multiscribe Reverse Transcriptase (Thermo Fisher Scientific, Waltham, MA, USA). Quantitative reverse transcription PCR was performed with SsoFast EvaGreen Supermix on a Bio-Rad CFX96 Real-Time System (Bio-Rad, Hercules, CA, USA). The primers used in this study are listed in Table S1.

## ROS assays

Reactive oxygen species (ROS) assays were performed as previously described (Pruitt et al., 2015; Wei et al., 2016). Fully expanded leaves from the same plants used for gene expression assays were cut into two mm$^2$ pieces and floated in water overnight. Four leaf pieces were pooled and treated with water, 500 nM RaxX21-Y, or 500 nM RaxX21-sY. Each treatment was replicated three times per experiment. Chemiluminescence was recorded in a high-sensitivity TriStar plate reader (Berthold, Bad Wildbad, Baden-Württemberg, Germany).

## Yeast two-hybrid assays

Yeast two-hybrid assays were performed using the Matchmaker LexA two-hybrid system (Clontech, Mountain View, CA, USA) following the procedures described previously (Chen et al., 2014). The pLexA and pB42AD vectors were co-transformed into yeast pEGY48/p8op-lacZ (Clontech, Mountain View, CA, USA) using the Frozen-EZ yeast transformation II kit (Zymo Research, Irvine, CA, USA). To confirm the expression of LexA- and HA-fusion proteins, yeast co-transformed with the appropriate vectors were grown in glucose free SD liquid media overnight with shaking at 30 °C. Cells were pelleted at 3,000 rpm for 5 min, resuspended in 4× SDS loading buffer, and incubated for 10 min at 95 °C to denature proteins.

## Purification of recombinant proteins

GST-XA21JK and its variants were expressed in BL21(DE3) pLysS cells (Novagen, Madison, WI, USA). Cultures were induced with 0.3 mM IPTG at 23 °C for 16 h. After 16 h incubation, cultures were pelleted and resuspended in Cell Wash Buffer (50 mM MOPS, pH 7.5, 150 mM NaCl). A total of 250 µg/ml lysozyme was added to the cell suspension and incubated at room temperature with shaking for 30 min. After 30 min, the cell suspension was cooled on ice for at least 30 min. 20× protease inhibitor (10 µM leupeptin, 0.5 mM AEBSF, one mM Benzamidine, two mM Caproic acid, and 10 µM E64) was added to the cell suspension and cells were sonicated for 4.5 min each on ice. The cellular debris was pelleted for 15 min at 4 °C. The supernatant was transferred into fresh, ice-cooled 50 ml tubes. GST-fusion proteins were purified using Glutathione–Agarose gel (Sigma-Aldrich, St. Louis, MO, USA) following the manufacturer's protocol. After elution from the beads, the protein solution was dialyzed against a 1,000× volume of buffer containing 20 mM Mops, pH 7.5, and one mM DTT.

His-Nus-fusion proteins were expressed in BL21 cells. Extraction and purification were described in previous studies (*Liu et al., 2002*; *Schwessinger et al., 2011*). Cultures were induced with 0.5 mM IPTG at 16 °C overnight. Cells were pelleted and suspended in lysis buffer (25 mM Tris, 500 mM NaCl pH 8.0, 2.5 mM EDTA, one mM PMSF, and one mg/ml lysozyme). The bacterial suspension was placed on a shaker for 1 h at room temperature. Bacterial suspension was pelleted and resuspended on ice in extraction buffer (lysis buffer, 40 mM imidazole, one mM $MgCl_2$, one mM DTT, and ~one µl Nuclease). The bacterial suspension was sonicated for 4 min followed by 10 min shaking on ice. Samples were pelleted and the protein was enriched using a 0.5 µM filter. The fusion proteins were purified using HISTRAP $^{TM}$FF (GE Healthcare, Chicago, IL, USA) according to the manufacturer's protocols. After elution, the concentration of fusion proteins was equalized using 10% glycerol solution and stored at −20 °C until usage.

## In vitro kinase assays

Kinase autophosphorylation assays were performed as described previously (*Liu et al., 2002*). In brief, purified fusion proteins were incubated in kinase buffer (50 mM Tris·HCl, pH 7.5, two mM MgCl2, two mM $MnCl_2$, and one mM DTT), three µM cold ATP, and five µCi [γ-$^{32}$P]-ATP at 30 °C for 30 min with gentle shaking. The reactions were stopped by adding 4× SDS loading buffer and incubating at 80 °C for 10 min. Proteins were separated by 10% SDS–PAGE and the phosphorylation of fusion proteins was analyzed by autoradiography. Densitometry measurements were carried out using imageJ 1.8.0 (*Schneider, Rasband & Eliceiri, 2012*).

## Immunoblotting

Protein extracts from bacteria, yeast, and rice plants were subjected to separation by SDS–PAGE, transferred to polyvinyl difluoride membranes, and immunoblot analysis was performed as described previously (*Chen et al., 2014*). Epitope-tagged proteins were incubated with the appropriate primary antibodies: anti-GST (Genscript, Nanjing, China), anti-phosphotyrosine (Genscript, Nanjing, China), anti-GFP (Santa Cruz Biotech, Dallas, TX, USA), anti-LexA (Clontech, Mountain View, CA, USA), or anti-HA (Covance, Princeton, NJ, USA). Detection of anti-GST and anti-phosphotyrosine primary antibodies was performed using an Alexa Fluor 680-conjugated secondary antibody (Rockland Immunochemicals, Gilbertsville, PA, USA), and imaged using an Odyssey® Infrared (LI-COR Biosciences, Lincoln, NE, USA) imaging system. All other primary antibodies were incubated with horseradish peroxidase-conjugated anti-mouse (Santa Cruz Biotech, Dallas, TX, USA), or horseradish peroxidase-conjugated anti-rabbit (GE Healthcare, Chicago, IL, USA) secondary antibodies. Chemiluminescence substrate (Thermo Fisher Scientific, Waltham, MA, USA) was used to detect the horseradish peroxidase-conjugated secondary antibodies by exposure using film or Bio-Rad ChemiDoc$^{TM}$ imaging system. Protein quantification was done using Coomassie Brilliant Blue (CBB) staining. Densitometry measurements were carried out using imageJ 1.8.0 (*Schneider, Rasband & Eliceiri, 2012*).

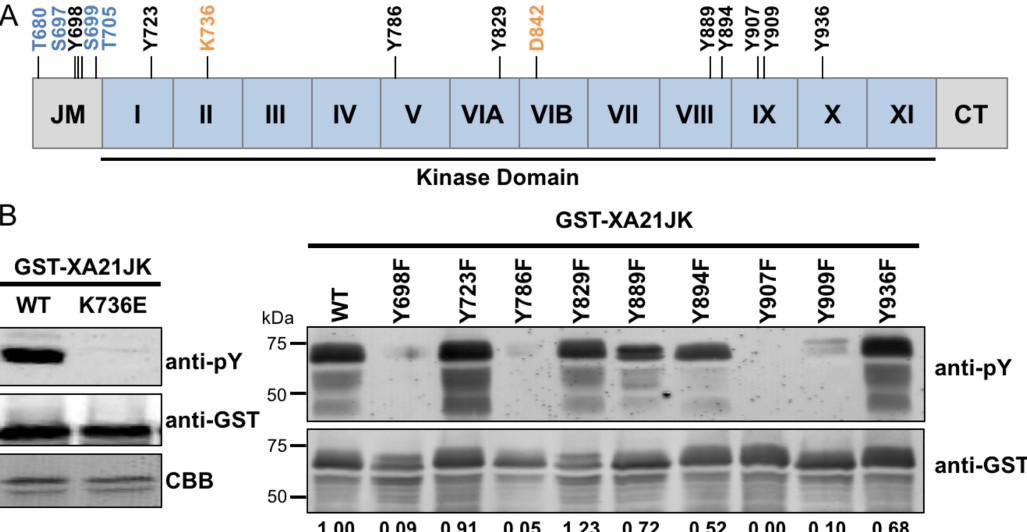

**Figure 1 XA21 is autophosphorylated on tyrosine residues in vitro.** (A) Representation of the XA21 juxtamembrane (JM), kinase subdomains (I–XI), and C-terminal region (CT) (XA21JK). Full-length XA21 has N-terminal leucine-rich repeats and a transmembrane domain that are not depicted. The positions of the nine XA21JK tyrosine residues (Y) are indicated in black. The conserved lysine (K) required for kinase activity and aspartate (D) required for phospho-transfer are labeled in orange. Previously studied XA21 JM Threonine (T) and Serine (S) residues are depicted in blue. (B) Immunoblots of wildtype (WT) GST-tagged XA21 (GST-XA21JK) and GST-XA21JK with lysine modified to glutamine (GST-XA21JK$^{K736E}$) and WT GST-XA21JK and variants with individual tyrosine residues modified to phenylalanine (Y698F, Y723F, Y786F, Y829F, Y889F, Y894F, Y907F, Y909F, Y936F). Tyrosine-specific phosphorylation was detected by immunoblotting with anti-phosphotyrosine antibodies (anti-pY). Equal loading of proteins was confirmed by immunoblotting with an anti-GST antibody and/or Coomassie Brilliant Blue (CBB) staining of the membrane. Densitometry measurements shown below the panel represent the ratio between phosphotyrosine and GST signals normalized to WT GST-XA21JK. The above experiments were performed twice with similar results.

# RESULTS

## Identification of XA21 residues required for in vitro tyrosine autophosphorylation

We previously demonstrated that the *E. coli*–expressed juxtamembrane and kinase domain of XA21 (referred to here as XA21JK, represented in Fig. 1A and Fig. S1), fused with MBP or GST, autophosphorylates on multiple serine and threonine residues (*Liu et al., 2002*). More recent studies have demonstrated that other plant LRR-RLKs are also capable of autophosphorylation on tyrosine residues (*Macho et al., 2014*; *Oh et al., 2009*). This was observed by assaying phosphorylation by immunoblotting using anti-phosphotyrosine specific antibodies, which detect tyrosine phosphorylated proteins, but does not allow for the identification of specific Tyr-phosphorylated residues. We applied this technique to XA21JK, to determine if it has tyrosine phosphorylation activity that was previously undetected (*Liu et al., 2002*), and we observed tyrosine autophosphorylation on XA21JK but not the catalytically inactive variant XA21JK$^{K736E}$ (Fig. 1B). This result demonstrates that XA21 has tyrosine autophosphorylation activity. A total of nine tyrosine residues, Tyr$^{698}$, Tyr$^{723}$, Tyr$^{786}$, Tyr$^{829}$, Tyr$^{889}$, Tyr$^{894}$, Tyr$^{907}$, Tyr$^{909}$, Tyr$^{936}$, are located in the

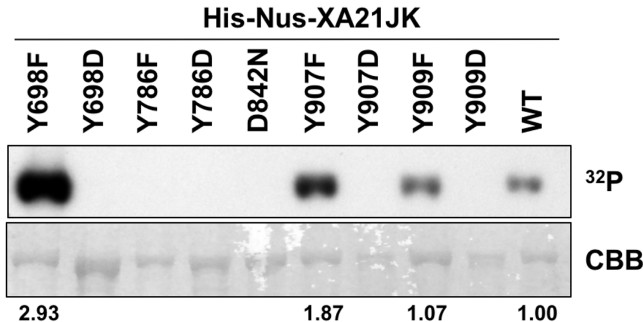

**Figure 2 Kinase autophosphorylation activity is lost in certain His-Nus-XA21JK variants.** Kinase autophosphorylation assay of purified *E. coli* expressed His-Nus-XA21JK wild-type (WT) and variant proteins (Y698F, Y698D, Y786F, Y786D, D842N, Y907F, Y907D, Y909F, Y909D). Kinase autophosphorylation was assessed by [$\gamma$-$^{32}$P]-ATP ($^{32}$P) incorporation (see Materials and Methods). The autoradiogram (top) and Coomassie Brilliant Blue (CBB) stained gel (bottom) of the same gel are shown. His-Nus-XA21JK$^{D842N}$ is catalytically inactive and is included as a negative control. Densitometry measurements shown below the panel represent the ratio between $^{32}$P signal and CBB stain normalized to WT GST-XA21JK. This experiment was performed twice using independent protein purifications with similar results.

XA21 JK domains (Fig. 1A; Fig. S1). To determine which of the nine tyrosine residues were potential sites of autophosphorylation, each tyrosine residue was substituted with phenylalanine, which lacks the hydroxyl group required for phosphate transfer. Phosphorylation of the tyrosine variants was monitored by immunoblotting. We found that XA21JK$^{Y698F}$, XA21JK$^{Y786F}$, XA21JK$^{Y907F}$, and XA21JK$^{Y909F}$ had reduced tyrosine autophosphorylation signals compared with wildtype, suggesting that Tyr$^{698}$, Tyr$^{786}$, Tyr$^{907}$, and Tyr$^{909}$ are potential sites of tyrosine autophosphorylation in wildtype XA21. Tyrosine phosphorylation also appeared slightly reduced in XA21JK$^{Y889F}$, XA21JK$^{Y894F}$, and XA21JK$^{Y936F}$, whereas the other tyrosine directed mutants maintained a signal similar to wildtype XA21JK (Fig. 1B). Because the reduced signal associated with Tyr$^{698}$, Tyr$^{786}$, Tyr$^{907}$, and Tyr$^{909}$ were most clearly demonstrable, these four residues were selected for further study.

## In vitro purified XA21JK$^{Y698F}$, XA21JK$^{Y907F}$, and XA21JK$^{Y909F}$ variants are catalytically active

To determine if Tyr$^{698}$, Tyr$^{786}$, Tyr$^{907}$, or Tyr$^{909}$ are required for XA21 catalytic activity, we expressed the XA21JK, XA21JK$^{D842N}$, XA21JK$^{YF}$, and XA21JK$^{YD}$ variants in *E. coli*. We then purified the proteins and incubated each protein in the presence of radiolabeled ATP ([$\gamma$-$^{32}$P]-ATP). Autophosphorylation of XA21JK and the mutant variants was determined by the ability of each protein to incorporate radiolabeled phosphate. These experiments indicated that XA21JK possesses kinase activity, whereas XA21JK$^{D842N}$ does not (Fig. 2), confirming a previous report (Chen et al., 2014). Autophosphorylation was also observed in XA21JK$^{Y698F}$, XA21JK$^{Y907F}$, and XA21JK$^{Y909F}$ variants, suggesting that these tyrosine residues are not required for in vitro XA21 catalytic activity (Fig. 2). Furthermore, XA21JK$^{Y907F}$ was slightly more active and XA21JK$^{Y698F}$ was hyperactive compared with XA21JK (Fig. 2). In contrast, autophosphorylation was not

detected in XA21JK$^{Y786F}$, and all four XA21JK$^{YD}$ variants, suggesting that aspartic acid substitutions at these residues disrupt in vitro XA21 kinase activity (Fig. 2), and that the hydroxyl group of Tyr$^{786}$ is essential for in vitro kinase activity (Fig. 2).

Serine and threonine residues within the juxtamembrane domain of XA21 regulate protein stability (*Xu et al., 2006*), kinase activity (*Chen et al., 2010a*), and protein–protein interactions (*Chen et al., 2010a*; *Park et al., 2008*). We previously demonstrated that autophosphorylation is reduced in XA21JK$^{S697A}$ and XA21JK$^{S697D}$ variants and enhanced in XA21JK$^{T680A}$ and XA21JK$^{S699A}$ proteins (*Chen et al., 2010a*). We observed similar levels of autophosphorylation in XA21JK$^{S686A/T688A/S689A}$ and XA21JK. In contrast, XA21JK$^{T705A}$ and XA21JK$^{T705E}$ lost autophosphorylation activity (*Chen et al., 2010a*). The contribution of these residues in tyrosine autophosphorylation was not tested.

To determine whether these previously characterized XA21 variants alter XA21 tyrosine autophosphorylation patterns, we carried out immunoblotting with anti-pY antibodies on in vitro recombinant proteins purified from *E. coli*. For these experiments, we tested XA21JK variants that had Ser$^{697}$, Thr$^{705}$, Thr$^{680}$, or Ser$^{699}$, replaced with alanine, and XA21JK variants that had Ser$^{697}$ and Thr$^{705}$ substituted with aspartate or glutamate (*Chen et al., 2010a*). We found that the XA21JK$^{S697D}$ variant displayed tyrosine autophosphorylation similar to wildtype, whereas XA21JK$^{S697A}$, XA21JK$^{T705A}$, XA21JK$^{T705E}$, and XA21JK$^{S699A}$ variants each displayed reduced tyrosine autophosphorylation (Fig. S2). Because tyrosine phosphorylation was reduced in XA21JK$^{S697A}$ but not XA21JK$^{S697D}$, phosphorylation of Ser$^{697}$ may facilitate autophosphorylation of one or more tyrosine residues in vitro. We were unsuccessful in expressing XA21JK$^{T680A}$, so the contribution of Thr$^{680}$ toward tyrosine phosphorylation was not determined (Fig. S2).

## Transgenic rice plants expressing XA21$^{YD}$-GFP variants are susceptible to *Xoo*

To determine if Tyr$^{698}$, Tyr$^{786}$, Tyr$^{907}$, and Tyr$^{909}$ affect the immune function of XA21, we generated Kitaake rice plants expressing a C-terminal GFP-tagged XA21 construct (XA21-GFP) and XA21-GFP constructs carrying a YD or YF substitution at one of these residues. All constructs were driven by the maize ubiquitin promoter. The GFP tag was previously reported to not affect XA21 function (*Park et al., 2013*). At least 10 independent rice plants were generated for XA21-GFP and for each XA21-GFP variant. Expression of XA21-GFP and the XA21-GFP variants were assessed in the $T_0$ generation by immunoblot analysis using anti-GFP antibodies. The $T_0$ transgenic lines with confirmed XA21 expression were inoculated with *Xoo* (Fig. S3). XA21$^{YD}$-GFP variants were as susceptible to *Xoo* as the Kitaake control plants, while XA21$^{YF}$-GFP variants displayed reduced lesion development.

To further characterize the effect of the mutations, at least two independent lines per tyrosine substitution with confirmed expression of the XA21-GFP variants (XA21$^{YF}$-GFP lines 698F-1, 698F-2, Y698F-9, 786F-4, 786F-5, 907F-7, 907F-8, 909F-9, and 909F-11 and XA21$^{YD}$-GFP lines 698D-5, 698D-6, 786D-1, 786D-2, 907D-11, 907D-12, 909D-11, and 909D-16) were advanced to the next generation ($T_1$). In the $T_1$ generation,

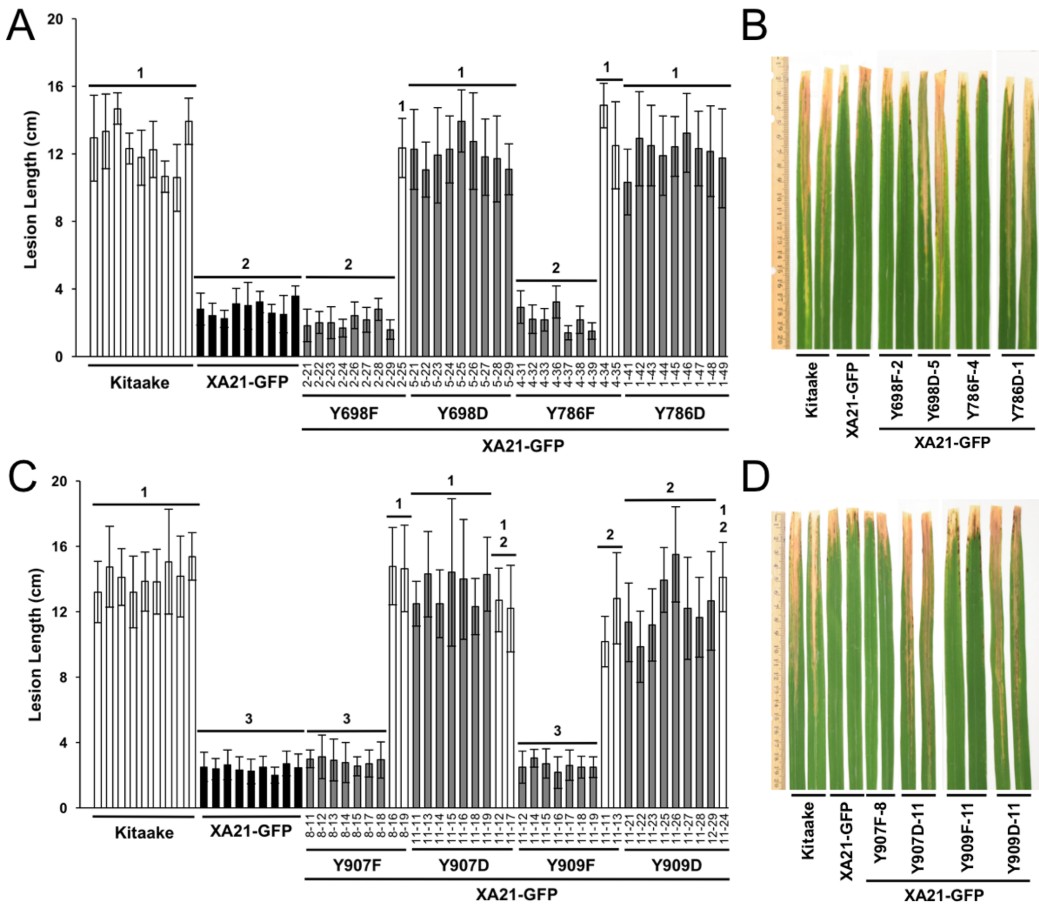

**Figure 3 Rice plants expressing XA21$^{YD}$-GFP variants are susceptible to *Xoo*.** Lesion length of Kitaake, XA21-GFP, and T$_1$ generation mutant plants 14 days after inoculation with *Xoo*. Bars indicate the mean lesion length and standard deviation of a single plant with four to 11 leaves inoculated with *Xoo* OD = 0.5. Different numbers indicate a significant difference in lesion length ($P < 0.05$, Kruskal–Wallis test, Dunn's post-hoc test with Benjamini–Hochberg correction). Gray bars indicate the presence of the XA21-GFP construct as assessed by PCR (Primers in Table S1). White bars indicate null-segregants. Shown is one representative transgenic line per tyrosine substitution. At least one additional independent transgenic line per tyrosine substitution was inoculated with similar results (Fig. S4). This experiment was performed three times with similar results. Two representative leaves per transgenic line were photographed at the time of lesion measurement (B and D). (A & B) and (C & D) were inoculated on separate dates.

XA21$^{YF}$-GFP variants, XA21$^{YD}$-GFP variants, XA21-GFP (progeny from homozygous line 5B-5-4-3-1), and Kitaake rice plants were inoculated with *Xoo*. We found that the XA21-GFP control and each of the XA21$^{YF}$-GFP variants were resistant to *Xoo*, with limited lesion development. In contrast, each of the XA21$^{YD}$-GFP variants and null-segregants developed water-soaked lesions similar to those observed on the control Kitaake plants. One representative line per tyrosine substitution is presented in Fig. 3. Lesion lengths for at least one additional independent line per tyrosine substitution is presented in Fig. S4. These results indicate that substitution of tyrosine with aspartic acid, a change that may cause structural alterations to XA21 and is predicted to mimic the negative charge of a phosphorylated tyrosine, disrupts XA21-mediated immunity.

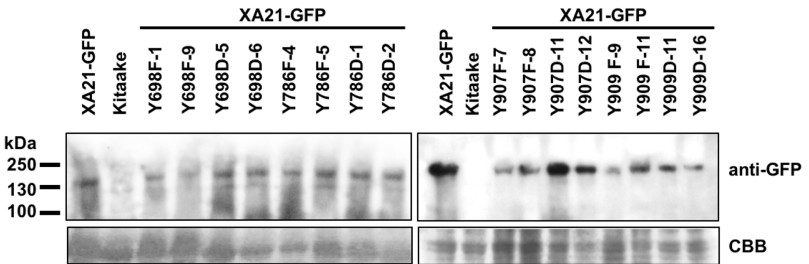

**Figure 4  XA21-GFP proteins are expressed in rice leaves.** Immunoblotting of XA21-GFP and mutant variants was performed using anti-GFP antibodies (top panel). A signal corresponding to the expected molecular weight of XA21-GFP was detected at ~170 kDa in XA21-GFP and two independently-transformed lines per tyrosine substitution. No signal was detected in Kitaake, which lacks Xa21. Equal loading of proteins was confirmed by Coomassie Brilliant Blue (CBB) staining of the membrane after immunoblotting (lower panel). This experiment was repeated three times with similar results.

This result is consistent with the observation that XA21JK$^{YD}$ variants are incapable of autophosphorylation in *E. coli*.

To confirm the expression of XA21-GFP and each of the XA21-GFP variants in the $T_1$ plants, protein was extracted from two independent PCR-validated transgenic lines per mutant variant and the Kitaake control line. Similar amounts of protein with the predicted molecular weight of 170 kDa corresponding to XA21-GFP were detected in XA21-GFP, XA21$^{YF}$-GFP (lines 698F-1, 698F-9, 786F-4, 786F-5, 907F-7, 907F-8, 909F-9, and 909F-11) and XA21$^{YD}$-GFP (lines 698D-5, 698D-6, 786D-1, 786D-2, 907D-11, 907D-12, 909D-11, and 909D-16) plants. We did not detect a similar sized band in the control Kitaake rice plants that lack XA21-GFP (Fig. 4). These results suggest that although XA21$^{YD}$-GFP variants may have structural changes to their protein structure compared to wildtype, the susceptibility of XA21$^{YD}$-GFP plants to *Xoo* was not caused by XA21 protein instability.

## XA21$^{YD}$-GFP variants are unresponsive to RaxX21-sY peptide treatment

Sulfated RaxX peptides (RaxX21-sY) are sufficient to activate XA21-mediated defense responses in the absence of *Xoo*. ROS production and activation of the XA21 genetic markers *Os04g10010* and *PR10b* are two hallmarks of perception of RaxX21-sY by XA21 (*Pruitt et al., 2015*; *Thomas et al., 2016*). Sulfation of RaxX is necessary for recognition, and the non-sulfated RaxX21-Y peptide does not initiate the ROS burst or XA21 marker gene activation (*Pruitt et al., 2015*). Because XA21$^{YD}$-GFP plants are susceptible to *Xoo*, we hypothesized that tissue from these plants would not respond to RaxX21-sY peptide treatment. To test this hypothesis, we treated leaf clippings from Kitaake, XA21-GFP, and XA21$^{YD}$-GFP variants (lines Y698D-5-14, Y786D-1-2, Y907D-4-2, and Y909D-11-4) with RaxX21-sY, RaxX21-Y, or water, and assessed their immune responses. We found that XA21-GFP leaf clippings displayed a ROS burst (Fig. 5A) and increased expression of *Os04g10010* and PR*10b* (Fig. 5B) following treatment with RaxX21-sY, but not RaxX21-Y or water. Notably, neither Kitaake, which lacks XA21,

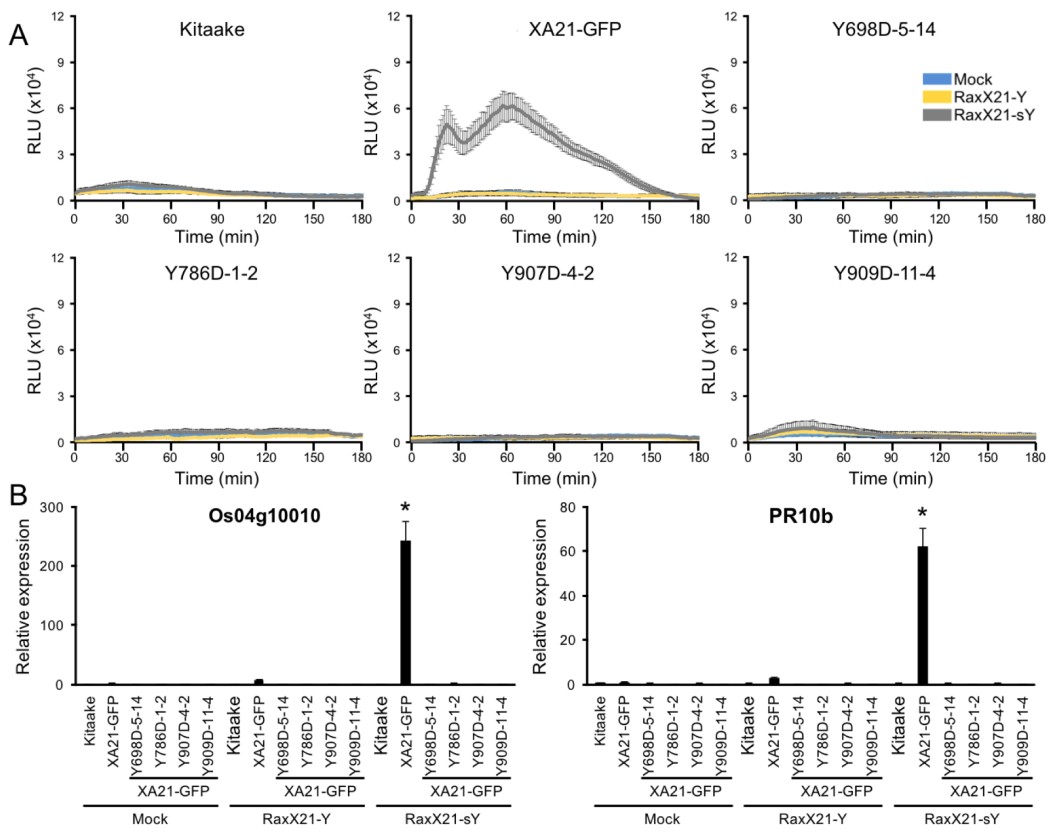

**Figure 5 XA21$^{YD}$-GFP variants do not respond to sulfated RaxX.** (A) ROS production in rice leaves of Kitaake, XA21-GFP, and XA21$^{YD}$-GFP variants 6 h after treatment with water (mock, blue), one μM RaxX21-Y (yellow) or one μM RaxX21-sY (gray). Data points depict the mean ± standard error ($n = 3$). Each panel depicts a different plant background from a single experiment. RLU, relative light units. (B) Gene expression of *Os04g10010* (left) and *PR10b* (*Os12g36850*, right) in rice leaves of Kitaake, XA21-GFP, and XA21$^{YD}$-GFP variants 6 h after treatment with water (mock), 500 nM RaxX21-Y, or 500 nM RaxX21-sY. Bars depict the means ± standard deviation ($n = 3$) of expression level normalized to mock-treated XA21-GFP. "*"Indicates a significant difference in gene expression compared to the mock treatment ($P < 0.05$, ANOVA, Tukey–HSD). Similar results were obtained from one additional independent transgenic line per tyrosine variant.

or the four XA21$^{YD}$-GFP variants responded to any of the treatments (Fig. 5). These results indicate that the XA21$^{YD}$-GFP variants fail to activate XA21-mediated defense responses in response to RaxX21-sY, consistent with the observed susceptibility of the XA21$^{YD}$-GFP variants to *Xoo*.

We next hypothesized that XA21$^{YF}$-GFP variants, which are resistant to *Xoo*, would react to RaxX21-sY. To test this hypothesis, leaf clippings of XA21-GFP, Kitaake, and XA21$^{YF}$-GFP variants (lines Y698F-2-2, Y786F-5-8, Y907F-7-2, and Y909F-6-5) were treated with RaxX21-sY, RaxX21-Y, or water. Immune responses were assessed as described for the XA21$^{YD}$-GFP variants above. We observed that rice plants expressing XA21-GFP and each of the four XA21$^{YF}$-GFP variants were responsive to RaxX21-sY treatment, but not RaxX21-Y or water (Fig. 6). This observation supports the conclusion that XA21$^{YF}$-GFP variants are able to perceive RaxX21-sY and initiate an XA21-mediated immune response. Notably, Y698F-2-2 displayed a fourfold higher RaxX-induced

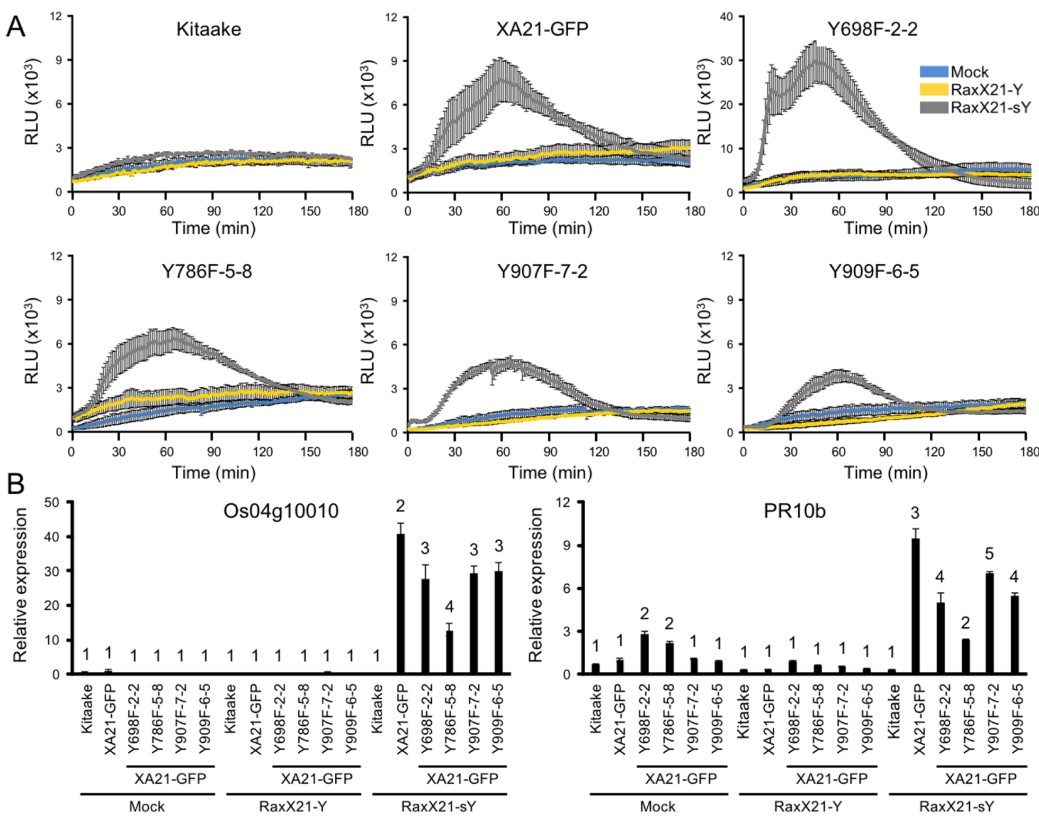

**Figure 6 XA21^YF-GFP variants respond to sulfated RaxX.** (A) ROS production in rice leaves of Kitaake, XA21-GFP, and XA21^YF-GFP variants 6 h after treatment with water (mock, blue), one µM RaxX21-Y (yellow) or one µM RaxX21-sY (gray). Data points depict the mean ± standard error ($n$ = 3). Each panel depicts a different plant background from a single experiment. RLU, relative light units. Note the $Y$-axis scale for Y698F-2-2 differs from the other plant lines. (B) Gene expression of *Os04g10010* (left) and *PR10b* (*Os12g36850*, right) in rice leaves of Kitaake, XA21-GFP, and XA21^YF-GFP variants 6 h after treatment with water (mock), 500 nM RaxX21-Y, or 500 nM RaxX21-sY. Bars depict the means ± standard deviation ($n$ = 3) of expression level normalized to mock-treated XA21-GFP. Different numbers indicate a significant difference in gene expression ($P < 0.05$, ANOVA, Tukey–HSD). The above experiments were performed twice with similar results.

ROS (Fig. 6), which may be associated with the hyperactive autophosphorylation which was observed in XA21JK^Y698F in vitro (Fig. 2). In contrast, Y907F-7-2, displayed reduced ROS compared with wildtype (Fig. 6), while XA21JK^Y907F had only a slight increase in autophosphorylation in vitro (Fig. 2). Similar to Y907F-7-2, Y909F-6-5 also showed reduced ROS compared to wildtype (Fig. 6). Collectively, these results suggest that each tyrosine residue has a different impact on signaling in vivo.

## XA21JK^YF proteins maintain interaction with XB3, XB15, and OsSERK2 in yeast

To determine if XA21 catalytic activity or tyrosine residues regulate protein–protein interactions, we tested the interaction of LexA-XA21JK, LexA-XA21JK^D842N, and the LexA-XA21JK^YF and LexA-XA21JK^YD variants with HA-XB3, an E3-ubiquitin ligase, HA-XB15, a protein phosphatase, and the co-receptor HA-OsSERK2JMK in yeast (Fig. 7).

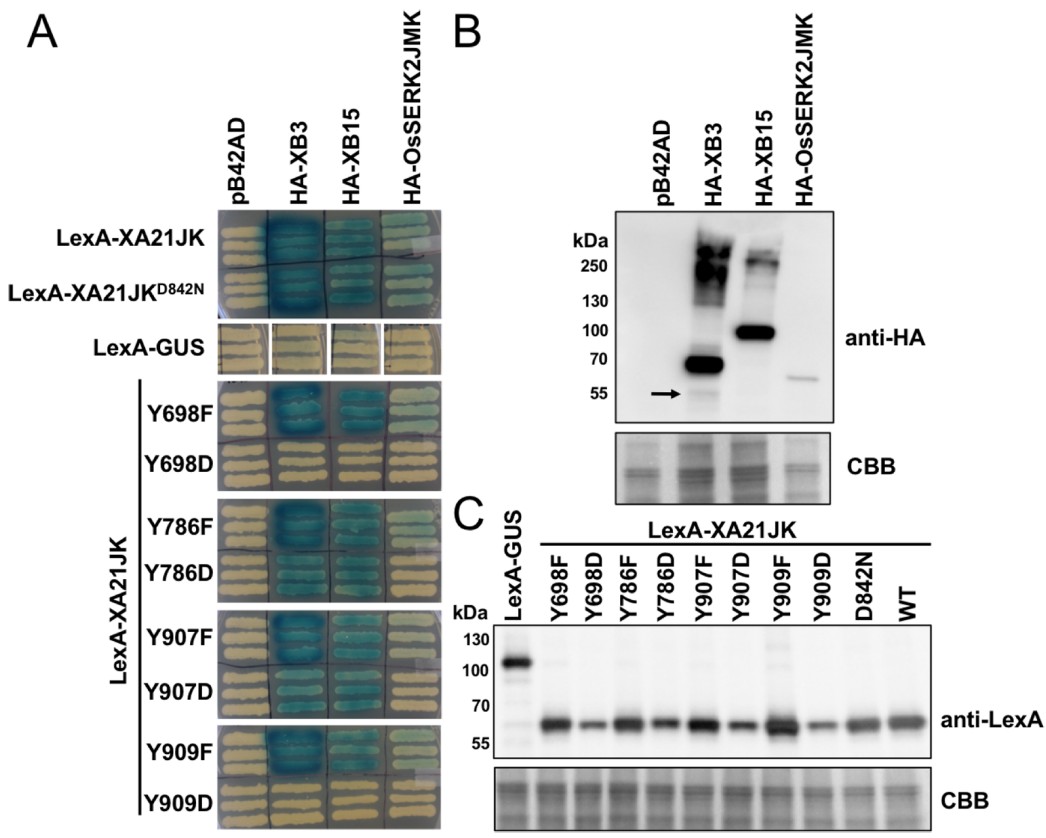

**Figure 7 XA21JK^YD variants fail to interact with known XA21 binding proteins.** (A) The interaction between wildtype (WT) LexA-XA21JK and mutant variants (fused to the yeast GAL4 binding domain) and HA-tagged XB3, XB15, and juxtamembrane and kinase domains of OsSERK2 (OsSERK2JMK) (fused to the yeast GAL4 activation domain) were tested. Blue color indicates nuclear interaction between the two co-expressed proteins. (B) HA-XB3, HA-XB15, and HA-OsSERK2JMK fusion proteins are expressed in yeast. Anti-HA primary antibodies detected HA-fused XB3, XB15, or OsSERK2JMK protein (upper panel). The predicted sizes of HA-XB3, HA-XB15, and HA-OsSERK2JMK are ~52, ~80, and ~65 kDa, respectively. Arrow indicates the band corresponding to the predicted size of HA-XB3. (C) WT Lex-A-XA21JK and mutant fusion proteins are expressed in yeast. Anti-LexA primary antibodies detected LexA-fused proteins (upper panel). LexA-GUS was used to confirm the absence of auto-activity in XB3, XB15, or OsSERK2JMK. The predicted size of LexA-GUS is ~110 kDa. The expected band size of LexA-XA21JK is ~65 kDa. Equal loading of proteins in (B) and (C) was confirmed by Coomassie staining (CBB) the membrane after immunoblotting (lower panels). The above experiments were repeated twice with similar results.

As previously demonstrated, LexA-XA21JK interacts with HA-XB3, HA-XB15, and HA-OsSERK2JMK (*Chen et al., 2014*; *Park et al., 2008*; *Wang et al., 2006*). LexA-XA21JK^D842N also maintains interaction with HA-XB3, HA-XB15, and HA-OsSERK2JMK, while no interaction was detected between any of the recombinant proteins and the empty vector controls, LexA-GUS or pB42AD (Fig. 7). This result indicates that XA21 catalytic activity is not required for interaction with XB3, XB15, or OsSERK2.

LexA-XA21JK^YF variants also maintain interaction with HA-XB3, HA-XB15, and HA-OsSERK2JMK (Fig. 7), indicating that these four tyrosine residues are not required for their interactions. However, the interaction with HA-OsSERK2JMK was disrupted in all

four LexA-XA21JK$^{YD}$ variants, suggesting that aspartic acid caused a structural change compared with LexA-XA21JK, or the negative charge of aspartic acid at these four sites disrupt the interaction with HA-OsSERK2JMK in yeast. The interaction with HA-XB3 and HA-XB15 was also disrupted in LexA-XA21JK$^{Y698D}$ and LexA-XA21JK$^{Y909D}$, but not LexA-XA21JK$^{Y786D}$ and LexA-XA21JK$^{Y907D}$ (Fig. 7).

## DISCUSSION

### The role of tyrosine phosphorylation in LRR-RLK-mediated responses

Plant LRR-RLKs regulate diverse signaling pathways and the kinase domain is typically at least partially critical for functionality. While many LRR-RLKs were first identified as serine/threonine protein kinases, recent examples have emerged of plant LRR-RLKs with dual specificity; that is, capable of serine/threonine and tyrosine phosphorylation. For example, tyrosine phosphorylation regulates two Arabidopsis LRR-RLKs, EFR (*Macho et al., 2014*) and BRI1 (*Oh et al., 2009*), and the LysM-RLK CERK1 (*Liu et al., 2018*). In EFR, phosphorylation of Tyr$^{836}$ is required for activation of immune activity (*Macho et al., 2014*; *Oh et al., 2009*). A role for the analogous tyrosine residue has also been identified in CERK1 (*Liu et al., 2018*; *Suzuki et al., 2018*), while BRI1 has a phenylalanine rather than a tyrosine at the corresponding residue (BRI1$^{F996}$, see Fig. S1). In contrast, phosphorylation of BRI1$^{Y956}$ is thought to inhibit kinase activity (*Oh, Clouse & Huber, 2012*). Tyrosine phosphorylation also regulates BRI1$^{Y831}$, as expression of BRI1$^{Y831F}$ enhances growth, and BRI1$^{Y831D}$ recovers wild-type leaf size and flowering time in *bri1-5* mutant plants (*Oh et al., 2009*). These results indicate that tyrosine phosphorylation is an important mechanism for regulating plant LRR-RLKs.

Previous work with EFR and CERK1 show that individual tyrosine phosphorylation sites which alone do not contribute to overall kinase activity in vitro are indispensable for signaling (*Liu et al., 2018*; *Macho et al., 2014*; *Suzuki et al., 2016*, *2018*). In this study, we observed that XA21$^{Y829F}$ does not reduce the detectable levels of autophosphorylated tyrosine in vitro, as was observed with EFR$^{Y836F}$ (*Macho et al., 2014*), and was not selected for further characterization. Thus, it would be very interesting to perform an in planta mutagenesis analysis of XA21$^{Y829}$ in order to compare regulatory mechanisms of XA21 and EFR, given that signaling components for XA21- and EFR-mediated immunity are known to be shared (*Chen et al., 2014*; *Holton et al., 2015*; *Schwessinger et al., 2015*; *Thomas et al., 2018*).

While previous studies have focused primarily on the importance of tyrosine phosphorylation by studying the effect of tyrosine phosphorylation absence (using YF mutations), less attention has been given to studying the effect of tyrosine phosphorylation presence (using phosphomimetic YD mutations). This is in part because phosphorylation mimicking mutations do not perfectly mimic phosphorylated amino acids, and may cause steric alterations or structural changes (*Corbit et al., 2003*; *Liu et al., 2018*; *Paleologou et al., 2008*; *Suzuki et al., 2018*). In this study, we demonstrate that aspartic acid substitution of XA21 tyrosine residues Tyr$^{698}$, Tyr$^{786}$, Tyr$^{907}$, and Tyr$^{909}$, disrupt catalytic activity of XA21, interaction with essential signaling proteins, and XA21-mediated
immunity. Further studies are needed to assess if autophosphorylation of specific XA21 tyrosine residues occurs in vivo.

## The mechanism regulating the gatekeeper residue differs between kinases

In 1995, Hanks and Hunter identified a "gatekeeper" residue, located in kinase subdomain V, adjacent to the hinge that connects the N-and C-lobes. This residue is located immediately distal to the active site and regulates access to the ATP binding pocket (*Hanks & Hunter, 1995*; *Paul et al., 2014*). In plant kinases, tyrosine residues are highly conserved at the gatekeeper position. For example, a previous study found that 83% of aligned Arabidopsis RLK/RLCK family proteins contained tyrosine at the gatekeeper position (*Klaus-Heisen et al., 2011*). The gatekeeper tyrosine residue is present in EFR, BRI1, and XA21 (EFR$^{Y791}$, BRI1$^{Y956}$, and XA21JK$^{Y786}$) (Fig. S1). In contrast, another Arabidopsis LRR-RLK, FLS2, contains a leucine residue at the gatekeeper position, suggesting that the gatekeeper tyrosine residue is not a requirement for biological activity in plant LRR-RLKs.

Studies assessing the substitution of the gatekeeper tyrosine residue with phenylalanine have found contrasting effects on catalytic activity. For example, a phenylalanine substitution mutant of the conserved gatekeeper tyrosine in the Symbiosis Receptor Kinase from *Arachis hypogaea* (*Ah*SYMRK$^{Y670F}$) is catalytically active, suggesting it is not involved in phosphotransfer (*Samaddar et al., 2013*). In contrast, the lysin motif domain-containing receptor-like kinase-3 (LYK3$^{Y390F}$) in *Medicago truncatula* only maintains partial kinase activity (*Klaus-Heisen et al., 2011*), and autophosphorylation activity is completely abolished in BRI1$^{Y956F}$ (*Oh et al., 2009*). In this study, we identified XA21$^{Y786}$, the gatekeeper tyrosine residue, as one of three tyrosine residues in the XA21 kinase domain predicted to function in tyrosine phosphorylation. Like BRI1$^{Y956F}$, we found that XA21JK$^{Y786F}$ autophosphorylation is significantly reduced (Fig. 2), suggesting the gatekeeper tyrosine residue is important for catalytic activity in XA21.

Phosphorylation of the gatekeeper residues also has varying effects on biological activity between kinases. For example, BRI1$^{Y956F}$ is inactive in vivo, whereas *Ah*SYMRK$^{Y670F}$, *Mt*LYK3$^{Y390F}$, and XA21$^{Y786F}$-GFP are all biologically active. EFR$^{Y791F}$ is also able to initiate EFR-mediated immunity. However, catalytic activity of EFR$^{Y791F}$ has not been established (*Macho et al., 2014*). A follow up study in *Ah*SYMRK utilizing the phosphomimetic mutant *Ah*SYMRK$^{Y670E}$ suggested that phosphorylation of the gatekeeper residue is autoinhibitory, rather than essential for activity (*Paul et al., 2014*). Our study suggests that XA21 functions similarly, as XA21$^{Y786F}$-GFP is biologically active, while XA21$^{Y786D}$-GFP is not. Collectively, our work provides further evidence that the mechanism regulating the gatekeeper tyrosine residue differs between kinases and must be assessed on a case-by-case basis.

## XA21 catalytic activity is not required for interaction with OsSERK2

Interaction between Arabidopsis LRR-RLKs such as FLS2 and EFR and the co-receptor BAK1 are ligand dependent, occurring rapidly upon perception of their cognate bacterial

peptides (*Schulze et al., 2010*). While an active kinase is required for biological activity, it is not required for ligand-induced complex formation with BAK1 (*Schwessinger et al., 2011*). In contrast, catalytic activity of BRI1 is required for complex formation with BAK1 (*Wang et al., 2008*). In rice, the co-receptor OsSERK2 forms interactions with three different rice LRR-RLKs, XA21, XA3, and OsFLS2 (*Chen et al., 2014*). Unlike complex formation between BAK1 and the LRR-RLKs of Arabidopsis, XA21 forms a constitutive complex with OsSERK2 in the absence of bacterial elicitors (*Chen et al., 2014*). The XA21–OsSERK2 complex was suspected to be kinase activity dependent, as catalytically inactive XA21JK$^{K736E}$ fails to interact with OsSERK2JMK in yeast (*Chen et al., 2014*). However, in this study we identified an additional XA21 variant, XA21JK$^{D842N}$, which is catalytically inactive, yet maintains interaction with OsSERK2JMK in yeast, indicating that catalytic activity is not the sole determinant of XA21–OsSERK2 interaction. Alternatively, it is possible that the replacement of the positive charged lysine with the negative charge of glutamate in XA21JK$^{K736E}$ may interfere with the interaction between proteins, or be causing a structural change in the protein. Thus, we suggest that future claims regarding the necessity of catalytic activity on protein–protein interactions should ideally include the use of multiple catalytically inactive variants, as the impacts of individual variants can differ.

While it has been demonstrated that the mechanism controlling LRR-RLK and co-receptor interaction differs between rice and Arabidopsis, the mechanism controlling signaling initiation is currently unknown. Biological activity of FLS2 and EFR both require active kinases. Similarly, rice expressing XA21$^{K736E}$ are only partially resistant to *Xoo* (*Andaya & Ronald, 2003*). However, XA21 may differ from FLS2 and EFR, as rice expressing XA21$^{Y786F}$-GFP are resistant to *Xoo* (Fig. 3), suggesting that the ability of XA21 to autophosphorylate in vitro does not correlate with XA21-mediated resistance in planta. We hypothesize that OsSERK2 transphosphorylation of XA21$^{Y786F}$-GFP is sufficient to confer resistance to *Xoo*, whereas the interaction between OsSERK2 and XA21JK$^{K736E}$ is weaker (*Chen et al., 2014*), and thus XA21JK$^{K736E}$ is only sufficient to provide the partial resistance observed previously (*Andaya & Ronald, 2003*). In agreement with this hypothesis, it was demonstrated that OsSERK2JMK is able to transphosphorylate XA21JK$^{K736E}$ despite not being able to observe a direct interaction (*Chen et al., 2014*). However, we cannot rule out that XA21$^{Y786F}$-GFP has additional in vivo kinase activity that was not observable in vitro (Fig. 2). Collectively, these results suggest that the OsSERK2–XA21 interaction is required for full resistance to *Xoo*, but the interaction is not dependent on these four specific tyrosine residues. Furthermore, the mechanism regulating activation of XA21 differs from previously studied LRR-RLKs including BRI1, EFR, or FLS2.

## CONCLUSIONS

While many plant LRR-RLKs were first identified as serine/threonine protein kinases, recent examples have emerged of plant LRR-RLKs capable of serine/threonine and tyrosine phosphorylation. In this study, we determined that XA21 is capable of performing in vitro tyrosine autophosphorylation. However, the four tyrosine residues

which we analyzed are not required for activation of XA21-mediated immune responses, or interaction with predicted XA21 signaling components including the co-receptor OsSERK2.

## ACKNOWLEDGEMENTS

We thank Dr. Benjamin Schwessinger for helpful discussions and technical assistance in performing kinase autophosphorylation assays and Dr. Gitta Coaker for her supportive discussions of the manuscript and the application of statistical methods. We express our gratitude to Dr. Christopher Harvey for helping with the assembly of raw data in preparation for peer review.

### Funding

This work was supported by the National Institutes of Health (NIH, GM55962 and GM122968), Foundation for Food and Agriculture Research grant # 534683 and the National Science Foundation (NSF, IOS-0817738). Daniel F. Caddell was supported by the Monsanto Beachell-Borlaug International Scholars Program. The funders had no role in study design, data collection and analysis, decision to publish, or preparation of the manuscript.

### Grant Disclosures

The following grant information was disclosed by the authors:
National Institutes of Health (NIH): GM55962 and GM122968.
Foundation for Food and Agriculture Research grant: # 534683.
National Science Foundation (NSF): IOS-0817738.
Monsanto Beachell-Borlaug International Scholars Program.

### Competing Interests

Pamela C. Ronald is an Academic Editor for PeerJ.

### Author Contributions

- Daniel F. Caddell conceived and designed the experiments, performed the experiments, analyzed the data, prepared figures and/or tables, authored or reviewed drafts of the paper, approved the final draft.
- Tong Wei performed the experiments, analyzed the data, prepared figures and/or tables, authored or reviewed drafts of the paper, approved the final draft.
- Sweta Sharma performed the experiments, prepared figures and/or tables, authored or reviewed drafts of the paper, approved the final draft.
- Man-Ho Oh conceived and designed the experiments, performed the experiments, prepared figures and/or tables, authored or reviewed drafts of the paper, approved the final draft.
- Chang-Jin Park conceived and designed the experiments, authored or reviewed drafts of the paper, approved the final draft.

- Patrick Canlas conceived and designed the experiments, contributed reagents/materials/ analysis tools, authored or reviewed drafts of the paper, approved the final draft.
- Steven C. Huber conceived and designed the experiments, contributed reagents/materials/analysis tools, authored or reviewed drafts of the paper, approved the final draft.
- Pamela C. Ronald conceived and designed the experiments, contributed reagents/ materials/analysis tools, authored or reviewed drafts of the paper, approved the final draft.

## Data Availability

Raw data are available in the Supplemental Files.

## Supplemental Information

Supplemental information for this article can be found online at http://dx.doi.org/10.7717/ peerj.6074#supplemental-information.

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
