# Peer review of "Four tyrosine residues of the rice immune receptor XA21 are not required for interaction with the co-receptor OsSERK2 or resistance to Xanthomonas oryzae pv. oryzae"

_PeerJ, doi:10.7717/peerj.6074_

## Round 0.1 · original submission · Major Revisions

Dear Pam,

Thank you for submitting your manuscript to PeerJ. You will see from the reviewers’ evaluations that each of them, while having some positive comments about your paper, has concerns about the quality of the figures and the interpretation of the results. Based on the fairly extensive changes that are needed, I’m not able to accept your paper in its present form, however, I hope you will consider resubmitting your paper after the concerns have been addressed.

One issue is the lack of direct evidence that any of the four tyrosine residues selected for functional analysis is actually phosphorylated. I understand that would be a lot of extra work and if you are unable to include it I think it is necessary to more clearly address how you believe individual phenylalanine substitutions at these residues abolishes all tyrosine phosphorylation. Another major concern is the interpretation of the tyrosine-to-aspartate substitutions and the likelihood that these could be causing structural changes in the Xa21 protein rather than reflecting anything about phosphorylation. Again, I think this could be addressed at certain places in the Results section and then elaborated upon more in the Discussion. I would also ask you to consider adding data about the Xa21 Y829F variant since it could help address the phosphorylation versus structural perturbation issue. In addition to these points it is important that you address the comments about Figures 1C, 4, 5 and 6 and the other minor concerns of each of the reviewers.

Please let me know if you have any questions about these modifications.

Sincerely,

Greg Martin

Reviewer 1 ·

Basic reporting

Caddell et al describes four tyrosine residues in XA21 that are not required for interaction with OsSERK2 and are not required for resistance to Xoo. Of the residues they investigated, they found four of nine were unable to autophosphorylate in vitro, and none of these were required for resistance. When transgenically expressed in rice, mutation of these four residues still allowed detection of the RaxX21-sY peptide. They found that Y786F maintained interaction in yeast, showing that catalytic activity is not the sole determinant of XA21-OsSERK2 interaction.
While this paper is overall scientifically sound and well written, I do have some concerns that would improve the paper:

Basic Reporting

The language is generally clear, intro and background show context, and most references are included. Some figures need some work to improve clarity and/or description of experiment. Raw data is supplied.

Major Concerns
1. Figure 4: This figure is overexposed and/or has too much protein loaded so that you can’t see the expression. It’s a big smear without a particular band. The authors say no signal was detected in Kitaake, but with the exposure there definitely appears to be a band ~150 kDa. Without knowing what the size should be (since it’s never listed), it’s hard to say if this is a valid conclusion. The Coomassie stain also shows that the protein is not equally loaded. The right panel is also very dark

Minor concerns
2. Line 285: specify that the immunoblotting was done in in vitro purified recombinant protein (E. coli).
3. Line 325: what is the molecular weight of XA21-GFP? This isn’t said anywhere in the paper.
4. .
5. Figure 5: It is difficult to see the orange line and could be a problem especially for color blind people. Make the orange a thicker line or make the graphs bigger so it’s easier to see.
6. Line 367: what do XB3 and XB15 do? This is not in the introduction (other XB’s are in the introduction).
7. Figure 7: HA-XB3 should be 52kDa, but the gel is run too far to see this size (lowest ladder marker is 55 kDa). The protein looks like the wrong size (~70kDa or faint band at ~55 kDa). If it’s the faint band put an arrow to denote correct protein.
8. Figure 1: combine B and C into one panel
9. Line 428-429: FLAGELLIN SENSING 2
10. The amino acid annotation is not consistent throughout the paper, e.g. Leu vs leucine switches throughout the Discussion
11. Line 480: reference missing
12. Line 481: transphosphorylate

Experimental design

Original primary research is in the scope of the journal, the research question is fairly well defined, and it is stated how the research fills a knowledge gap. The methods are fairly well described but some details need clarified.

Major Concerns
1. Figure 6: the background levels for all panels are high and increase over time (compare to background/mock for Fig 5). Why is this? This may indicate contamination of the peptide.

Minor concerns
2. Figure 3: Lesion length was calculated on plants with four to eleven leaves. Did you mean “using” four to eleven leaves, or four to eleven leaves as the age/stage? If age, this is a big difference in age and disease response can vary greatly based on age. If not age, this is unclear. This wording is the same in supplemental Figure 4.
3. Figure 5 and 6: “ROS production in rice leaves (n=3)…” Is this three leaves? And are these from the same plant or three individual plants? In the methods they say they did four biological replicates. Is this the average of the four biological reps?

Validity of the findings

Data is fairly robust and statistically sound, but could use some additional experiments and/or clarification. Conclusions are fairly well stated but a few things should be addressed.

Major concerns
1. There are no experiments that look at the phosphorylation of the tyrosine residues in plants.
2. Lines 357-360: The comparison to wildtype can not be made unless all of these ROS assays were done in the same experiment/same graph. This does not look like the case since they all look like separate experiments. If this comparison is to be made, wildtype and the mutants should all be compared in a single experiment.

Minor concerns
3. Line 293-294. How do you conclude that “interaction among potential phosphosites, e.g., the phorphorylation of Ser697, may regulate tyrosine autophosphorylation.”? There is no evidence that shows this.
4. The authors never look at transphosphorylation in the paper.

Additional comments

no comment

Annotated reviews are not available for download in order to protect the identity of reviewers who chose to remain anonymous.

Reviewer 2 ·

Basic reporting

The writing and structure of the manuscript are clear and orthodox, and the manuscript includes all the relevant references.

Experimental design

The experimental design is appropriate, and the results constitute original primary research. The questions are well-defined, relevant and meaningful. The methods are appropriate and well described.

Validity of the findings

The data seems robust.

I have doubts regarding whether the western blots provided as loading control in specific figures really correspond to the experimental blots shown, since the shape of several bands is considerably different (e.g. Fig 1C). This should be clarified or substituted.

Additional comments

My main concern about this article relies on whether or not the phenotypes observed in several experiments are really associated to phosphorylation or catalytic activity. In several instances, the phenotypic results do not seem to correlate with the observations regarding phosphorylation, which would at least justify significant modifications to the manuscript to explain these observations. Examples are below:

- Based on the figure 2, the authors suggest that several tyrosine residues play important role in XA21 kinase activity, since their YD variants have abolished phosphorylation. However, their respective YF variants (except Y786F) keep autophosphorylation, suggesting that phosphorylation in these residues does not really play a role in XA21 kinase activity. Considering the important steric alterations caused by YD mutations, it seems likely that the XA21 YD mutations have an impact in the correct folding of the protein, rather than their phosphorylation being required for kinase activity.

- A similar scenario is suggested by the results shown in figure 7. The D842N kinase-inactive version maintains the interaction with the tested XA21 interactors; sames as the YF variants. However, these interactions are abolished by several YD mutations, suggesting a problem in protein folding rather than specific effects of phosphorylation in these residues.

- Other examples come from the use of the Y786F mutant. This mutation abolishes autophosphorylation. However, transgenic lines expressing this XA21 mutant are resistant to Xoo (as other YF mutations). However, in the same instance, all the YD variants fail to provide resistance. Similarly, in the Figure 6, the authors associate the higher ROS value in the Y698F mutant to its hyperactive autophosphorylation shown in the figure 2. However, again, the Y786F, which loses completely the autophosphorylation activity, shows a ROS value similar to WT XA21.

In addition to this, the authors mention in the discussion that they observed that the mutation XA21Y829F does not abolish tyrosine autophosphorylation, and was not selected for further characterization. They mention the interest of doing mutagenesis analysis of this residue (mutant that they already have). Considering the relevance of the equivalent tyrosine residue in other RLKs, the correlation of the results presented in this manuscript with a similar analysis of the Y829F seems essential to uncouple potential effects of mutation of Y residues in phosphorylation and in the general folding of the protein, as I mentioned above.

Reviewer 3 ·

Basic reporting

Cadell et al investigates the role of tyrosine phosphorylation in regulating the function of XA21, a rice immune receptor that recognizes a sulfated peptide from Xanthomonas oryzae and provides resistance against this bacterial pathogen. In this work the authors report that recombinant XA21 autophosphorylates on tyrosine residues in vitro. Using site-directed mutagenesis, nine Tyr to Phe substitutions were made within the juxtamembrane and kinase domains of XA21. Four of these substitutions completely abolished in vitro autophosphorylation of XA21 on Tyr. The role of these four Tyr residues on overall XA21 function was further investigated by testing Tyr to Phe mutants for in vitro 32P-ATP autophosphorylation, XA21-mediated response to Rax21-sY peptide, XA21-mediated resistance to Xanthomonas, and XA21 interaction with known interacting proteins including SERK2. None of the four Tyr to Phe substitutions tested had any detrimental effect on these phenotypes, with the exception of Y786F disrupting XA21 autophosphorylation. In contrast, individual Tyr-Asp substitutions at the same four Tyr residues disrupted XA21 autophosphorylation, XA21-mediated response to Rax2-sY peptide and resistance to Xoo. The main conclusion of the study is that phosphorylation of the four Tyr residues identified is not required for XA21-mediated resistance or XA21 binding to interacting signaling proteins.

The manuscript is clearly written with background material that is relevant to the research topic.

There are several issues with the quality of figures that should be addressed:

Fig. 1C. The bands shown in the anti-pY blot seem to be upside down relative to the bands shown on the blot within the corresponding raw data file. Also, the contrast of this image has been adjusted to the point where it is difficult to assess if there are quantitative differences between bands, an important consideration for interpreting these results (see concerns regarding interpretation of this blot below).

The quality of the blot shown in Fig 4 left panel is poor (particularly when looking at the raw data file for this blot), to the point that it is difficult to assess levels of XA21-GFP expression.

Several legends do not provide details regarding sample size and replication: Fig. 1, no information about replication of results; Fig 5, no sample size for (A) or (B); Fig 6, no sample size; Fig. 7, no information about replication for (A) or (B).

Experimental design

Methods are well-described and technically sound. The research question is clearly defined.

Validity of the findings

Major concerns:

1. Figure 1C shows that altering any of four Tyr residues to Phe completely eliminates detectable Tyr phosphorylation in vitro. Based on these results, the authors conclude on line 259 that these four Tyr residues are potential phosphorylation sites. However, if there is a single site of Tyr phosphorylation on XA21, I would expect that only one of the four substitutions would be necessary to completely eliminate signal on the blot. If there are multiple sites of Tyr phosphorylation, then quantitative decreases in signal should occur. Therefore, it is difficult to see how four distinct Tyr to Phe substitutions would each completely abolish Tyr phosphorylation. The simplest explanation I can think of is that the Tyr to Phe substitutions are causing structural changes in XA21 that prevent Tyr autophosphorylation from occurring. This possibility needs to be more clearly stated throughout the manuscript including the discussion.
Also, there is no direct evidence in this work that any of the four Tyr residues selected for functional analysis are actually phosphorylated (confirmation by mass spec would be necessary for this, and would greatly strengthen conclusions of this work). While the authors are careful in the final conclusion section not to refer to the four Tyr residues as bona fide phosphorylation sites, there are multiple places within the manuscript that seem to suggest that these residues are indeed phosphorylated. For example, line 42 of abstract states that “in vivo phosphorylation of the identified tyrosine residues is not required for XA21-mediated immunity…”. Although technically correct (Tyr to Phe substitutions would prevent phosphorylation IF they are in fact phosphorylation sites), readers could be misled by this statement into thinking that these residues are indeed phosphorylated in vivo. The title of the manuscript is also misleading for similar reasons, and should be changed to more accurately reflect conclusions that can be drawn from data– rather than “Phosphorylation of the rice immune receptor XA21 on four tyrosine residues” perhaps simply state that “Four tyrosine residues of XA21 are not required for…”.

Other misleading statements that should be corrected: line 116 “… in vivo phosphorylation of the identified tyrosine residues is not required…”; line 373 “…indicating that phosphorylation of these four residues is not required for their interactions.”; line 484 “…interaction is not dependent on XA21 phosphorylation of these specific tyrosine residues.” Also, remove or alter sentence “These results suggest…” starting on line 378.

2. Another issue with Fig 1C is the poor contrast adjustment of the blot. By my eye the bands for both Y889F and Y894F appear to be weaker than WT, suggesting that they could be sites of Tyr phosphorylation. The bands on this blot and on blots from replicate experiments should be quantified to help readers assess this possibility.

3. In previous work (Liu et al 2002) the authors performed phosphoamino acid analysis of autophosphorylated XA21 and did not find any evidence of phosphorylated tyrosine. The authors should comment on this discrepancy in results- is ability to detect phos-Tyr in current work due to greater sensitivity of antibody? Different in vitro conditions?

4. Aspartic acid is not structurally similar to phos-Tyr and is more likely to disrupt protein structure than act as a phosphomimic. Because of this, Tyr to Asp substitutions are not commonly made, and Tyr to Asp substitutions are not predicted to be phosphomimetic as stated in line 318. There is a strong possibility that phenotypes observed with Tyr to Asp variants in this work are due to structural perturbations of XA21 rather than effects due to phosphomimic, and this should be more clearly stated in results and discussion.


Minor concerns:

1. The authors report that Y698F in Fig. 2 is hyper-phosphorylated, but Y907F phosphorylation also seems increased relative to WT. Quantitation of bands in blot would be helpful.

2. Line 476-477: “rice expressing XA21Y786F-GFP are resistant to Xoo, suggesting that XA21 catalytic activity is not required for resistance” should be revised. Y786F does not autophosphorylate in vitro, but that does not mean it lacks catalytic activity. The only conclusion that can be made from this result is that the ability of XA21 to autophosphorylate in vitro does not correlate with XA21 function in resistance in planta.

---

## Round 0.2 · accepted · Accept

Your revisions have addressed the reviewers' concerns and I am pleased to accept your paper.

# Reviewer 1 ·

Basic reporting

Caddell et al has made significant improvements to the quality of the paper. Although no additional experiments were performed to address some of my concerns about the phosphorylation status of the variants in vivo or whether the variants can transphosphorylate, the authors have satisfactorily addressed these concerns in the text. The title of the manuscript has changed to more accurately reflect the scope of the study. Significant improvements have also been made to increase the clarity of the figures. While the brightness/contrast has been adjusted in Fig. 4 to improve the visibility of the Xa21-GFP bands, and the size of the expected protein is listed in the main text (~170 kDa), it would still be preferred to include this expected size in the figure legend since there is a second band visible ~130 kDa.

Experimental design

No additional experiments were performed since the last review, and clarifications have been made about the experimental design.

Validity of the findings

Appropriate clarifications have been made about the validity of the findings and the scope of the project.

Reviewer 3 ·

Basic reporting

no additional comments

Experimental design

no additional comments

Validity of the findings

no additional comments

Additional comments

The revised manuscript addresses all of my previous concerns.

A few minor edits:
Densitometry measurements in Fig 1B show Y936F phosphorylation is reduced, similar to Y889F and Y894F. Was this consistently observed across multiple experiments? If so, Y936F should be added to this statement on Line 265: “Tyrosine phosphorylation also appeared slightly reduced in XA21JKY889F and XA21JKY894F, whereas the other tyrosine directed mutants maintained a signal similar to wild-type XA21JK (Fig. 1B)”.

Please correct misplaced modifier on line 295: “we carried out immunoblotting of in vitro recombinant proteins purified from E. coli with anti-pY antibodies” to clarify that purification of E coli proteins was not done with anti-pY.